# A Novel Approach to Enhance the Regenerative Potential of Circulating Endothelial Progenitor Cells in Patients with End-Stage Kidney Disease

**DOI:** 10.3390/biomedicines10040883

**Published:** 2022-04-12

**Authors:** Amrilmaen Badawi, Osfred C. Jefferson, Brooke M. Huuskes, Sharon D. Ricardo, Peter G. Kerr, Chrishan S. Samuel, Padma Murthi

**Affiliations:** 1Cardiovascular Disease Program, Department of Pharmacology, Monash Biomedicine Discovery Institute, Monash University, Melbourne, VIC 3800, Australia; amrilbadawi@gmail.com (A.B.); ocjefferson310172@gmail.com (O.C.J.); sharon.ricardo@monash.edu (S.D.R.); 2Centre for Cardiovascular Biology and Disease Research, Department of Microbiology, Anatomy, Physiology and Pharmacology, La Trobe University, Melbourne, VIC 3086, Australia; b.huuskes@latrobe.edu.au; 3Department of Nephrology, Monash Medical Centre, Melbourne, VIC 3168, Australia; peter.kerr@monash.edu; 4Department of Biochemistry and Molecular Biology, University of Melbourne, Melbourne, VIC 3010, Australia; 5Department of Obstetrics and Gynaecology, University of Melbourne, Melbourne, VIC 3010, Australia

**Keywords:** endothelial progenitor cells, end-stage kidney disease, bone-marrow derived mesenchymal stem cells, relaxin, wound healing, angiogenesis, regeneration

## Abstract

Circulating bone marrow-derived endothelial progenitor cells (EPCs) facilitate vascular repair in several organs including the kidney but are progressively diminished in end-stage kidney disease (ESKD) patients, which correlates with cardiovascular outcomes and related mortality. We thus determined if enhancing the tissue-reparative effects of human bone marrow-derived mesenchymal stromal cells (BM-MSCs) with the vasculogenic effects of recombinant human relaxin (RLX) could promote EPC proliferation and function. CD34^+^ EPCs were isolated from the blood of healthy and ESKD patients, cultured until late EPCs had formed, then stimulated with BM-MSC-derived condition media (CM; 25%), RLX (1 or 10 ng/mL), or both treatments combined. Whilst RLX alone stimulated EPC proliferation, capillary tube formation and wound healing in vitro, these measures were more rapidly and markedly enhanced by the combined effects of BM-MSC-derived CM and RLX in EPCs derived from both healthy and ESKD patients. These findings have important clinical implications, having identified a novel combination therapy that can restore and enhance EPC number and function in ESKD patients.

## 1. Introduction

Chronic kidney disease (CKD) is a global health problem [1], defined as a reduction in glomerular filtration rate (GFR) ≤ 60 mL/min/1.73 m^2^, that often leads to end-stage (stage V) kidney disease (ESKD; defined as GFR ≤ 15 mL/min/1.73 m^2^) [2]. The pathological features of CKD include tubular atrophy associated with reductions in renal capillaries and podocytes, destruction of renal endothelial cells and the development of renal fibrosis [3,4]. The kidney endothelium is compromised during these processes, leading to impaired angiogenesis and kidney function [5]. Following kidney damage, bone marrow-derived endothelial progenitor cells (EPCs) are recruited to sites of injury to facilitate tissue repair [6]. EPCs play pivotal roles in the maturation and proliferation of kidney endothelial cells, vascular integrity and repair of the damaged endothelium [7]. However, circulating EPC numbers are progressively reduced in ESKD patients [7,8,9], which correlates with adverse cardiovascular outcomes [9,10] and long-term mortality [11].

Mesenchymal stromal cells (MSCs) have been reported to promote angiogenesis by stimulating EPC programming [12]. However, while MSCs have been evaluated in various clinical trials [13], their ability to facilitate kidney endothelial regeneration has not been investigated. Recent studies from our laboratory have identified the enhanced therapeutic potential of combining human bone marrow (BM)-derived MSCs with an anti-fibrotic agent, namely recombinant human (gene-2) relaxin (serelaxin; RLX [14]) in ameliorating kidney damage, inflammation and fibrosis in preclinical models of disease [15,16,17]. Whilst RLX alone can stimulate human blood-derived EPC proliferation and nitric oxide (NO) production in vitro, and vasculogenesis in mice in vivo [18], it remains unknown if its combined effects with BM-MSCs could accelerate and/or enhance EPC-derived angiogenesis and wound healing. This study thus, evaluated the therapeutic potential of combining RLX and BM-MSC-derived conditioned media (CM) on healthy versus ESKD patient-derived EPC proliferation, angiogenesis and wound healing in vitro.

## 2. Materials and Methods

### 2.1. EPC Isolation and Culture from Peripheral Blood

Upon receipt of a signed patient consent form, around 5–40 mL of blood from healthy male controls was collected from the Australian Red Cross Blood Service. In contrast, only ~5 mL maximum of blood of male ESKD patients was collected from the Monash Medical Centre due to their health condition. All blood samples were processed within 24 h of collection time. All blood samples were diluted with 1× Hank’s balanced salt solution (HBSS; Invitrogen, Carlsbad, CA, USA) to a total volume of 20 mL. Then, the blood was carefully layered over 15 mL of Ficoll-Paque Plus (GE Healthcare, Uppsala, Sweden) into a 50 mL Falcon tube (BD Biosciences, San Jose, CA, USA) and centrifuged at 400× *g* for 40 mins at room temperature to separate the buffy coat from other components by using a density gradient separation method as previously described [19,20]. Following the gradient separation, the upper layer containing plasma and platelets was removed by carefully inserting a serological pipette, leaving the buffy coat containing mononuclear cells from peripheral blood (PBMCs) undisturbed, consisting of endothelial progenitor cells (EPCs). The pellet was resuspended in 2 mL of Endothelial Cell Growth Media (EGM)-2 Microvascular (MV) Bullet Kit (product #CC-3162, Lonza, Hayward, CA, USA); supplemented with 5% fetal bovine serum (FBS), 0.04% hydrocortisone, 0.4% human fibroblast growth factor (hFGF), and 0.1% of vascular endothelial growth factor (VEGF), R3-insulin-like growth factor (IGF)-1, ascorbic acid, human epidermal growth factor (hEGF), and gentamicin sulfate/amphotericin (GA-1000), to culture isolated EPCs. The cells were counted using a Countess^TM^ Automated Cell Counter (Invitrogen, Carlsbad, CA, USA) before they were seeded onto fibronectin-coated culture plates/flasks.

The yield of EPC varied significantly between the uncomplicated control and ESKD patients. More specifically EPC yield was significantly lower in ESKD patient samples compared to control and this observation was consistent with previous studies reported in our laboratory [19]. EPCs isolated from healthy controls were directly seeded into T-75 flasks (1–10 × 10^7^ cells per sample), which were supplemented with 10 mL of pre-warmed endothelial growth media-2 (EGM-2). In contrast, due to the low yield of EPCs from the ESKD patient samples, the cells were seeded at a density of 1 × 10^6^ cells per well into 12 well plates, with the addition of pre-warmed EGM-2, giving a total volume of 2 mL per well. Due to the large variability in the yield of EPCs isolated from ESKD patients, cells were plated in one well of a 12 well plate if the cell yield was less than 1 × 10^6^ cells. The media was changed every second day until outgrowth endothelial cells (OECs, also known as late EPCs) were observed. These late EPCs were identified by their cobble-stone appearance under light microscopy (Olympus CK-X41, USA). A maximum period of 30 days was allowed for late EPCs to be detected, which were further cultured until ~80% confluency was reached. The cells were then passaged 1–2 times to obtain the desired number of cells for all functional assays including proliferation, tube formation and wound-healing assays as detailed below. Cellular morphological characteristics were further assessed by the ability of the EPC cultures in both control and ESKD patients to form a number of colonies (>50 cells/colony) using the colony forming unit (CFU) assay.

### 2.2. Preparation of Conditioned Media (CM) from Cultured BM-MSC

As the addition of BM-MSCs to EPC cultures would compromise the EPC-associated endpoints to be measured, it was decided that it would be better to analyze the effects of BM-MSC-derived CM on EPC function, as previously used for the analysis of other endpoints [19]. Briefly, BM-MSCs were grown in Alpha-Minimum Essential Media (α-MEM; Sigma-Aldrich, Castle Hill, NSW, Australia), supplemented with 16% fetal bovine serum (FBS) and 1% 200 mM l-glutamine, and 1% penicillin/streptomycin. Once the cells reached ~80% confluency, BM-MSCs were further sub-cultured and plated into 12-well plate at a density of ~0.5 × 10^6^ per well and maintained in culture for 24–48 h to ensure cellular attachment. To prepare conditioned media (CM) from BM-MSCs, briefly the cells were washed three times with pre-warmed 1 mL of 1× HBSS. Following the washing steps, 500 µL of serum and growth factor-free endothelial cell basal media (EBM; Product #CC-3121, Lonza, Hayward, CA, USA) was added to the cells, and maintained in culture at 5% CO_2_; 37 °C for a further 24 h. At the end of this incubation period, CM was then collected and centrifuged at 400× *g* for 10 min to remove any cellular debris. CM in 500 µL aliquots was stored at −80 °C until future use.

### 2.3. Functional Assays

#### 2.3.1. Treatment of Cultured Late EPCs

Once the late EPCs were harvested, functional assays were performed. All experiments were conducted in passages 3–6. Functional assays were performed to investigate the effect of (i) 25% BM-MSC derived conditioned media (25% CM) [20]; (ii) 1 [RLX-1] or 10 [RLX-10] ng/mL [18] RLX (representing circulating levels of H2 relaxin found in pregnant women [21]); or (iii) the combined effects of 25% CM and 1 or 10 ng/mL RLX; (iv) 20% FBS was used as a positive control on the proliferation and tube formation (angiogenic) potential of the late EPCs derived from control patients. Treatment groups were compared with untreated cells.

#### 2.3.2. Viability and Proliferation Assay

The viability and proliferative potential of the late EPCs was determined by using the 3-(4,5-dimethythiazol-2-yl)-2,5-diphenyl tetrazolium bromide (MTT) assay according to the manufacturer’s instructions (Sigma-Aldrich, Melbourne, VIC, Australia). Briefly, ~5 × 10^3^ cells per well were seeded in 96-well plates (BD Falcon, North Ryde, NSW, Australia), following which, the cells were treated with BM-MSC derived CM, RLX (10 ng/mL) or a combination of CM and RLX at 1 or 10 ng/mL for 24 h. At the end of the incubation period, 10 µL of MTT labelling reagent was added (0.5 mg/mL, Sigma-Aldrich, Melbourne, VIC, Australia) and incubated at 37 °C and 5% CO_2_ for a further 4 h. The cells were solubilised by the addition of 100 µL of the Solubilization solution (Sigma-Aldrich, Melbourne, VIC, Australia) and the proliferation potential of the late EPCs was determined by measuring the absorbance of each sample at 590 nm using a Micro-plate reader (Bio-strategy, Balwyn North, VIC, Australia) and analysed with SoftMax Pro software for Windows OS (Ver. 7.3, Molecular Devices LLC, CA, USA). Each of the treatment groups was performed and analysed in six replicates.

#### 2.3.3. Tube Formation Assay

The effects of the various treatments to promote the angiogenic potential of the late EPCs, using a tube formation assay were determined using the µ-angiogenesis assay (Ibidi, Fitchburg, WI, USA). Briefly, 10 µL of growth factor-reduced (GFR) Matrigel (Product #356231, Corning, Tewksbury, MA, USA) was added into the inner well of the µ-Slide of the angiogenesis assay kit (Ibidi, Fitchburg, WI, USA). Confluent cultures of the late EPCs were seeded onto the coated cells and treated with BM-MSC derived CM, RLX (10 ng/mL), or a combination of CM and RLX at 10 ng/mL. A total volume of 50 µL of cell suspension containing ~1 × 10^4^ cells was applied to each well. Images were immediately captured and labelled as the 0-hour time-point. The ability of the cells under various treatments to form tubes was observed and images were captured at 2–24 h post-cell seeding using a light microscope. The number of tubes, tube length, and the number of branching points were determined by using the ImageJ software.

#### 2.3.4. Wound Healing Assay

The effects of the various treatments on the regenerative potential of the late EPCs were determined using a wound healing assay. To perform the assay, ~1 × 10^4^ cells were suspended in 50 µL growth media and seeded into 2 well-silicone in 35 mm dish for wound healing assay (Ibidi Inc., Fitchburg, WI, USA). Artificial wounds within the dishes were created by gently removing the silicone wells using a sterile tweezer (gap distance = 500 ± 100µm). The dishes were replenished with 2 mL 50% EGM-2, with the addition of various treatments including BM-MSC derived CM, RLX (10 ng/mL), or a combination of CM and RLX at 1 or 10 ng/mL. The number of cells that migrated to close the wound was assessed by using light microscopy. Photomicrographs of the migrated cells were captured at 0–24 h post-treatment. The percentage of gap closure area was determined using ImageJ software.

#### 2.3.5. Image Analyses

All image analyses for the functional assays were performed using ImageJ for MacOS (Ver. 1.8, National Institute of Health, Bethesda, MD, USA). For the tube formation potential of the late EPCs, the images were analysed using the angiogenesis assay plugin and four different parameters were recorded including total length, number of meshes, number of junctions, and number of branches. For the wound healing assay, the wound closure area was analysed by manually marking cell-free area at each of the time-points. All measurements were performed at least 3 times to control for variation.

#### 2.3.6. Immunofluorescence 

Relaxin Family Peptide Receptor 1 (RXFP1; the cognate RLX receptor) protein localisation in cultured late EPCs was determined using immunofluorescence staining. Firstly, ~1 × 10^5^ cells grown on a 13 mm glass coverslip (coated with poly-d-lysine) in a 12-well plate were fixed in 4% PFA for 15 min at room temperature. The fixed cells were blocked for non-specific protein binding using 1% Bovine serum albumin (BSA) for 60 min at room temperature. At the end of the incubation period, the cells were washed in PBS for 5 min (3 times) and incubated with a rabbit polyclonal RXFP1 antibody (aa 609-624; A 9227; 1:2000; Immunodiagnostik AG, Bensheim, Germany) in 0.01% BSA overnight at 4 °C. The cells were washed with PBS and incubated with goat-anti rabbit Alexa-fluor 594 secondary antibody (1:500; Invitrogen, Scoresby, Victoria, Australia) in 0.01% BSA for 2 h at room temperature. Nuclear counterstaining was performed by adding 40 µL of DAPI in Vectashield mounting medium (Vector Laboratories, Burlingame, CA, USA) and coverslipped. Immunoreactive RXFP1 protein was visualised using a fluorescence microscope at ×40 magnification (Olympus BX51) and the images were merged using ImageJ software.

### 2.4. Statistical Analyses

All data is shown as the mean ± SEM and all statistical analyses were performed using GraphPad Prism^TM^ (Ver. 9; GraphPad Software Inc., San Diego, CA, USA). A one way-ANOVA was used to compare the effect of various treatments on proliferation (MTT assay) and angiogenic (tube formation assay) potential of the late EPCs, followed by a Tukey’s post-hoc test to allow for multiple comparisons between the treatment groups. A repeated measure- two way-ANOVA was used to compare the effect of the various treatments on wound closure by the late EPCs over time, followed by Tukey’s post-hoc test to compare each of the treatment groups. A *p*-value of less than 0.05 was considered statistically significant.

## 3. Results

### 3.1. Patient Demography

The control and ESKD patient subjects included in this study are listed in Table 1. The demographic parameters of the ESKD patients including body height and weight, dialysis vintage, disease etiology, and alcohol consumption were recorded and collected from the medical records stored by Monash Health, Clayton, Victoria, Australia. 

Whilst late EPC cultures from all *n* = 9 healthy patients were successfully obtained (from a yield of ~1.5 ± 0.3 × 10^6^ cells/mL of blood), data were acquired from *n* = 6–7 separate patient-derived cultures per treatment group to complete the various assays conducted. However, only five EPC cultures out of the *n* = 11 ESKD patient-derived blood samples matured into late EPCs (from a yield of ~2.2 ± 0.3 × 10^5^ cells/mL of blood), from which data were acquired from *n* = 3–5 separate patient-derived cultures per treatment group. Additionally, the number of EPC colonies derived from ESKD patients (4.3 ± 1.9; *p* < 0.01 vs. EPC colonies derived from patients) was significantly lower than those obtained from healthy patients (19.3 ± 8.6; *n* = 4 samples per group).

### 3.2. Functional Analysis of EPCs Derived from Control Patients

Late EPCs obtained and cultured from healthy males expressed the RXFP1 receptor on their cell surface, as shown in Figure 1A. Late EPCs demonstrated a significant increase in proliferative capacity (by ~43%) after 24 h (Figure 1B); EPC-induced tube formation (as determined by tube length (by ~45%) and numbers of branches, junctions and meshes (by ~60–90%) after 4 h (Figure 1C) and EPC-induced wound healing (by ~60%) after 8 h (Figure 1D), compared to untreated cells when stimulated by 10 ng/mL RLX (all *p* < 0.05 vs. unstimulated cells), but not by 25% BM-MSC-CM or 1 ng/mL RLX.

Strikingly, the combined effects of 25% BM-MSC-CM and 10 ng/ml RLX further enhanced EPC proliferation (by ~1.3-fold; *p* < 0.001 vs. 25% BM-MSC-CM or 10 ng/mL RLX alone; Figure 1B), EPC-induced tube formation (including tube length (by ~65%) and the number of branches, junctions and meshes (by ~1.1–2.4-fold; Figure 1C) over the same time-period and EPC-induced wound healing (by ~87–88%) after 4–8 h (*p* < 0.01 vs. 25% BM-MSC-CM after 8 h; Figure 1D).

Comparatively, 20% fetal bovine serum (positive control) stimulated EPC proliferation (by ~60%) after 24 h (*p* < 0.001 vs. unstimulated cells; Figure 1B) but failed to stimulate EPC-induced capillary tube formation (Figure 1C). Based on these findings from healthy patient-derived EPCs, the effects of 25% BM-MSC-CM and/or 10 ng/mL RLX was then evaluated on ESKD patient-derived cells.

### 3.3. Functional Analyses of EPC’s Derived from ESKD Patients

As previously described, only five EPC cultures from the *n* = 11 ESKD patient-derived blood samples matured into late EPCs, from which data were acquired from *n* = 3–5 separate patient-derived cultures/treatment group. These ESKD patient-derived EPCs still expressed RXFP1 (suggesting that they could also respond to RLX treatment; Figure 2A) but had reduced viability and proliferative capacity (by ~70%; *p* < 0.01 vs. untreated cells derived from normal patients) compared to their counterparts isolated and grown from normal patients (Figure 2B). Interestingly, the addition of 20% FBS (positive control) and all treatments evaluated, stimulated EPC proliferation (by ~1.2–1.6-fold) after 24 h, almost back to levels that were measured from untreated EPCs from normal patients (all no different to untreated cells derived from normal patients; Figure 2B). 10 ng/mL RLX alone promoted EPC-mediated tube length, number of junctions and number of meshes after 4 h (all by ~1–2.5-fold), whereas all three measures in addition to number of branches were further increased by the combined treatment (all by ~3.1–4.2-fold; all *p* < 0.05 vs. 25% BM-MSC-CM or 10 ng/mL RLX alone; Figure 2C). EPC-induced wound healing in vitro from ESKD patients took longer to achieve compared to that from healthy patients but was more rapidly stimulated by 10 ng/mL RLX (by ~54%) or the combination therapy (by ~63%) to almost complete closure after 24 h (Figure 2D).

## 4. Discussion

Previous studies have reported that the number of EPCs are decreased in ESKD patients, which has correlated to a significant increase of disease-induced mortality [19,22]. It was also reported that long-term haemodialysis, which is considered as a major therapy for ESKD, could also contribute to increasing this ESKD-induced loss of EPCs in patients [23]. However, other studies have also reported that haemodialysis of ESKD patients failed to restore the ability of EPCs alone to proliferate and migrate in vitro compared to EPCs derived healthy patients [8,23]. These combined findings provided the rationale for why agents that could improve EPC proliferation and function are urgently required. As expanding EPC yield for allogenic therapeutics has proven to being somewhat difficult and time consuming, our study proposed a novel treatment by combining BM-MSC-derived CM and RLX to enhance EPC proliferation and the functional capabilities of increased EPCs in vitro, with a major focus on providing a proof-of-concept study for the direct clinical translation in modulating the regenerative capacity of EPCs in vivo. For the first time, this study demonstrated the therapeutic potential of combining BM-MSC-CM with RLX as a means of promoting the viability, proliferative and regenerative capacity of human EPCs.

In this study, we demonstrated the successful isolation and characterisation of early and late EPCs from the blood of healthy and ESKD patients that were collected from a small volume of blood (around 5–40 mL). Most previous studies have used around 50 mL of peripheral blood as starting volume to isolate EPCs [24,25,26]. Some studies did not state the starting volume of blood required for EPC isolation [27,28]. Furthermore, late EPCs were identified according to their morphology as well as their highly proliferative capacity, and their colony forming ability, from 20 mL of peripheral blood [29], suggesting that a minimum of 20 mL of peripheral blood was required to obtain EPCs that could proliferate. In this study, although the sample numbers for ESKD patients were from a small cohort compared to control subjects, with a starting volume of ~5 mL of peripheral blood, we identified that EPC yields were not only low but could not always undergo progression of differentiation into late EPC colonies. Furthermore, early EPCs were observed from healthy controls in days 6–10 of culture, while the late EPCs appeared after 14–28 days in culture. These observations were consistent with previous reports [25]. Several factors could contribute to the differentiation or transformation of PBMCs into early and late EPCs.

Whilst BM-MSCs alone can induce reparative effects in the kidney via their anti-inflammatory, anti-apoptotic, antioxidant, pro-angiogenic effects, and stimulation of EPCs [12], the therapeutic efficacy of these stem cells is severely compromised in chronic disease settings [30]. BM-MSCs have also been shown to have greater renoprotective effects in many different experimental models and are currently being evaluated in various clinical trials [31]. Fewer studies though have investigated the angiogenic role of MSCs in fibrotic models of kidney disease. One study reported on the impact of MSC on the vasculature of the kidney, in which the conditioned media from BM-MSC enriched with growth factors, such as VEGF, HGF, and IGF-1 was able to enhance aortic EC growth and proliferation [32]. Furthermore, a study in a pre-clinical model of Alport disease (a genetic condition characterised by kidney microvascular abnormalities and functional failure), observed that mice treated with MSC-CM expressed a high level of VEGFR mRNA in the kidneys, but not in the circulation, and also enhanced renal EC proliferation in vitro [33].

A previous study found that cytokines and growth factors within the CM of umbilical cord-MSC significantly increased the number of tubes and the percentage of wound recovery of human umbilical vascular endothelial cells (HUVECs) in vitro [34]. The mechanism underlying this property was identified via the increased expression of the cytokines SDF-1-CXCR4 and MCP-1, which enhanced the rate of migration of HUVECs to close the wound area generated in the wound-healing assays conducted [34]. However, given our previous studies which demonstrated that 100% BM-MSC-CM alone could significantly stimulate measures of EPC proliferation, capillary tube formation, angiogenesis and wound healing [20], whereas the titration of BM-MSC-CM down to 25% diminished these effects [20], the current study employed the use of 25% BM-MSC-CM so that the potential synergistic effects of 25% CM and RLX could be optimally determined.

RLX demonstrates its angiogenic potential by inducing VEGF and basic fibroblast growth factor (bFGF) at wound sites in mice [35,36]. Additionally, RLX has been shown to increase human bone marrow-EPC-derived intracellular NO and migration in mice, suggesting that RLX could promote angiogenesis in vivo [18]. However, it was observed that RLX did not directly bind to ECs, but rather induced angiogenesis via its ability to promote growth factors that play a role in wound-healing [35,36]. On the contrary, this study demonstrated that the viability/proliferation, tube formation, and wound healing capacities of EPCs were significantly enhanced by the administration of 10 ng/mL of RLX, consistent with previous findings [18]. In addition, it was observed that both early and late EPCs expressed the RXFP1 receptor, which suggested that RLX would be able to directly bind to its receptor on EPCs to promote EPC proliferation and EPC-induced angiogenesis in vitro.

Previous studies from our laboratory have demonstrated that combining BM-MSCs and RLX more effectively attenuated or reversed organ inflammation and fibrosis in murine models of obstructive nephropathy [15], hypertensive CKD [16,17] or chronic allergic airways disease (AAD) [37] in vivo. The findings of this study have now extended those findings to demonstrate that the combined effects of BM-MSC-derived CM and RLX can enhance the viability and proliferation rate of human ESKD patient-derived EPCs in vitro (to levels measured in normal patients), and the ability of these cells to promote angiogenesis. This was demonstrated by the increased capacity of the combination therapy to promote EPC-induced capillary tube formation and wound healing compared to CM alone. In this context, it is possible that the NO-mediated vasculogenic effects of RLX [18], which resulted in the RLX (at 10 ng/mL)-induced augmentation of EPC number and EPC-mediated wound healing, were synergistically enhanced by the EPC-promoting effects of angiogenic mediators (such as VEGF and bFGF) that are secreted by BM-MSC-CM [12,38] and stimulated by RLX [35,36]. Despite the ex vivo nature of this study and the low number of ESKD patient-derived EPC cultures analysed, the synergistic benefits of the combination therapy evaluated was clearly evident.

## 5. Conclusions

As expanding EPC yield from allogeneic therapeutics has proven to be difficult and time consuming, this novel therapy might hold great promise as a treatment for ESKD sufferers. Overall, the findings from this study are of high clinical significance, particularly for ESKD patients, in which a loss of circulating EPC numbers contributes to impaired growth and angiogenesis [7,8,9,39], adverse cardiovascular outcomes [9,10] and related mortality [11]. Furthermore, as ESKD patients often die from vascular diseases (atherosclerosis, stroke, peripheral vascular disease) [40], this combination therapy may provide a novel means of slowing the progression of vascular disease in these patients. Moreover, developing a new therapy that can retard the progression of CKDs will save the health care system billions of dollars in renal replacement therapy. As both MSCs and RLX are already being assessed clinically, novel strategies incorporating this combination therapy can be fast-tracked for human trials.

## Figures and Tables

**Figure 1 biomedicines-10-00883-f001:**
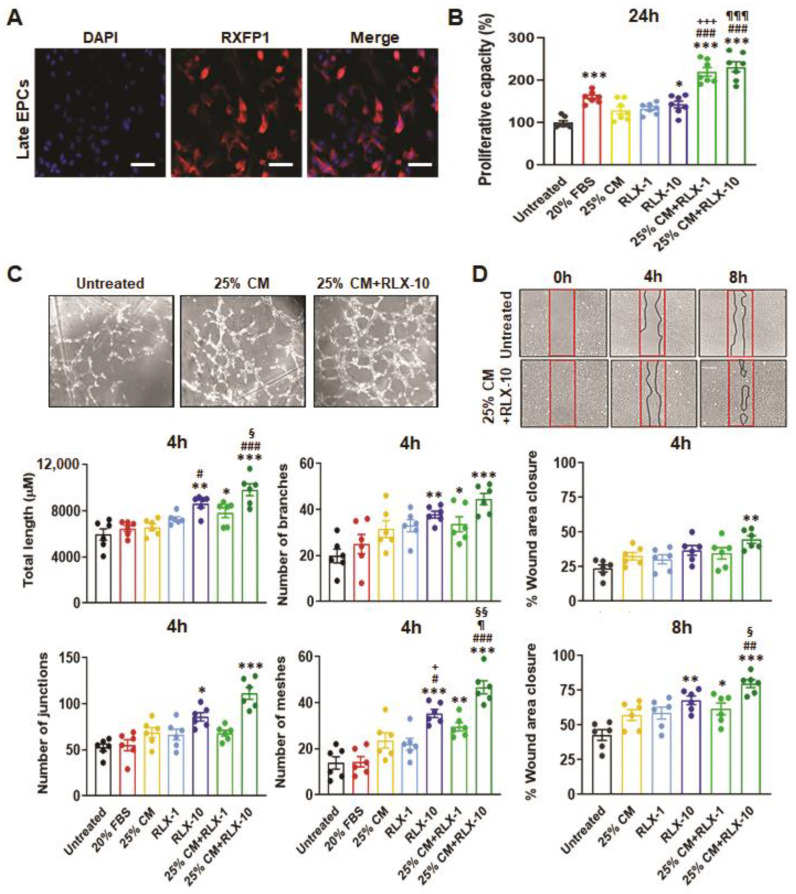
The effects of BM-MSC-derived CM and/or RLX on healthy patient-derived EPC viability/proliferation, angiogenesis and wound healing. CD34^+^ late EPCs from the blood of healthy male patients were found to express the cognate RLX receptor, RXFP1; scale bar = 50 µm (**A**). EPC proliferation after 24 hours (24 h) from *n* = 7 patient-derived EPCs per group (**B**); EPC-mediated tube formation (determined from total tube length, number of branches, number of junctions and number of meshes) after 4 h from *n* = 6 patient-derived EPCs per group (**C**) and EPC-mediated wound healing after 4 h and 8 h from *n* = 6 patient-derived EPCs per group (**D**), was determined in response to 20% fetal bovine serum (FBS; positive control), 25% BM-MSC-CM (25% CM), RLX [1 ng/mL (RLX-1) or 10 ng/mL (RLX-10)] or the combined effects of 25% CM and RLX-1 or RLX-10. Complete wound closure was achieved by untreated EPCs by 24 h. Representative images of capillary tube formation from untreated EPCs and EPCs treated with 25% CM alone or in combination with RLX-10 (**C**) and of wound closure from untreated EPCs and those treated with 25% CM+RLX-10 (**D**) are shown. The coloured circles in each of the columns of each bar graph represent individual data points from each group. Data were analysed using a one-way ANOVA followed by a Tukey’s post-hoc test to allow for multiple comparisons between groups. * *p* < 0.05, ** *p* < 0.01, *** *p* < 0.001 vs. untreated cells; ^#^ *p* < 0.05, ^##^ *p* < 0.01, ^###^ *p* < 0.001 vs. 25% CM-treated cells, ^+^ *p* < 0.05, ^+++^ *p* < 0.001 vs. RLX-1-treated cells; ^¶^ *p* < 0.05, ^¶¶¶^ *p* < 0.001 vs. RLX-10-treated cells; ^§^ *p* < 0.05, ^§§^ *p* < 0.01 vs. 25% CM+RLX-1-treated cells.

**Figure 2 biomedicines-10-00883-f002:**
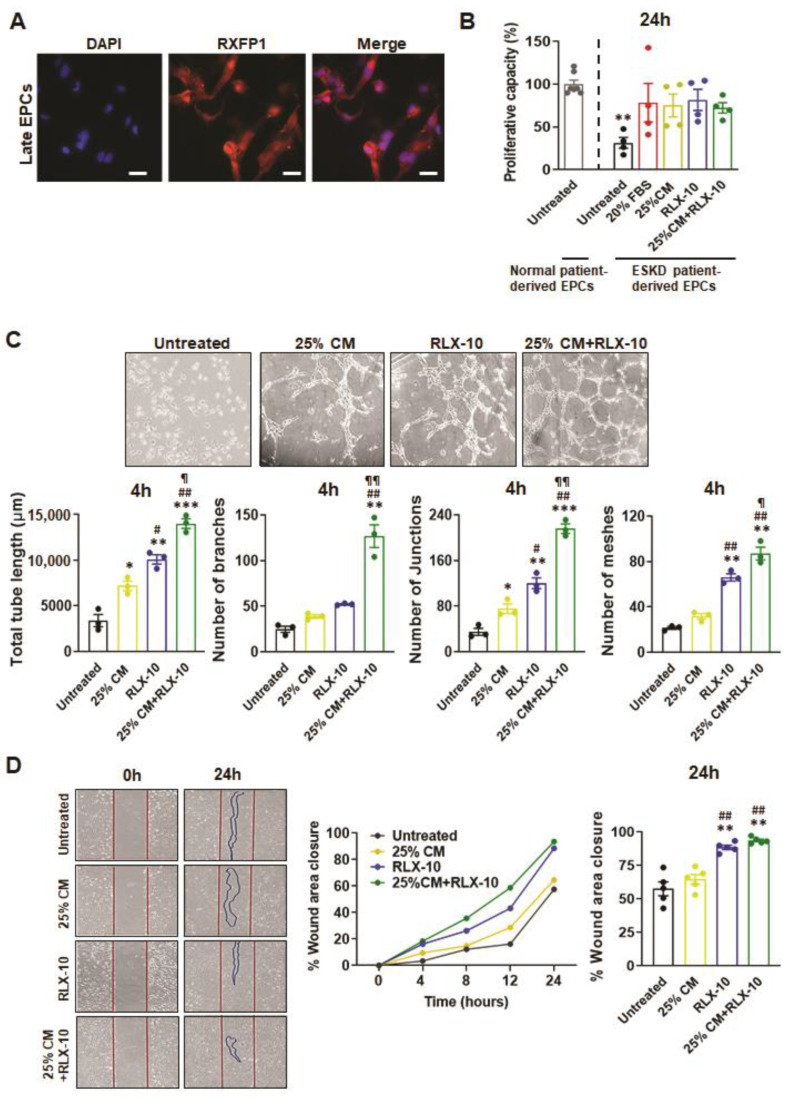
The effects of BM-MSC-derived CM and/or RLX on stage V ESKD patient-derived EPC viability/proliferation, angiogenesis and wound healing. Late EPCs derived from the blood of male ESKD patients were also found to express RXFP1; scale bar = 50 µm (**A**). EPC proliferation after 24 hours (24 h) was determined from *n* = 4 patient-derived EPCs per group in response to 20% FBS (positive control), 25% BM-MSC-CM (25% CM), RLX [10 ng/mL (RLX-10)] or the combined effects of 25% CM+RLX-10. Data from untreated EPCs derived from healthy male patients (presented in Figure 1B) are also included for comparison (**B**). EPC-mediated tube formation (determined from total tube length, number of branches, number of junctions and number of meshes) after 4 h from *n* = 3 patient-derived EPCs per group (**C**) and EPC-mediated wound healing after 24 h from *n* = 5 patient-derived EPCs per group (**D**), was determined in response to 25% BM-MSC-CM (25% CM), RLX [10 ng/mL (RLX-10)] or the combined effects of 25% CM+RLX-10. Representative images of tube formation (**C**) and wound closure (**D**) from each of the four groups analysed are shown. The coloured circles in each of the columns of each bar graph represent individual data points from each group; while the cumulative % wound healing closure from each of the treatment groups studied over 24 h is also shown (**C**). * *p* < 0.05, ** *p* < 0.01, *** *p* < 0.001 vs. untreated cells; ^#^ *p* < 0.05, ^##^ *p* < 0.01 vs. 25%CM-treated cells; ^¶^ *p* < 0.05, ^¶¶^ *p* < 0.01 vs. RLX-10-treated cells.

**Table 1 biomedicines-10-00883-t001:** Demographics of healthy control and ESRD patients used in this study.

Parameter	Healthy Control	ESKD Patients
Sample size (*n*)	9	11
Age (years)	49 ± 5	57 ± 6
Body height (cm)	Not Recorded	182 ± 5
Body weight (kg)		85 ± 6
BMI		24.7 + 0.6
Dialysis vintage (months)	Not Applicable	2.3 ± 0.7
*Disease etiology:*		
Chronic glomerulonephritis (%)		91
Amyloidosis (%)		9

Values are presented as the mean + SEM for each group. BMI, body mass index; cm, centimeters; kg, kilograms; *n*, number.

## Data Availability

The data that support the findings of this study are also available from the corresponding authors upon reasonable request. Some data may not be available because of privacy or ethical restrictions.

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
