# Peer review of "A Novel Approach to Enhance the Regenerative Potential of Circulating Endothelial Progenitor Cells in Patients with End-Stage Kidney Disease"

_biomedicines, 2022, doi:10.3390/biomedicines10040883_

Round 1

Reviewer 1 Report

This study dealt with the critical effects of MSC/RLX on function of EPC that is impaired by donor disease. Considering the importance of circulating EPC in vasculogenesis, improvement of EPC activity is necessary in clinic. Author provided the data to support their hypothesis for combined effect of MSC-CM and RLX.

  1. Authors have cultured EPC and determined its type depending on culture period (Ref 23). Did author try to check stem cell marker expression? Although culture time is the decisive point to verify EPC type, cell-specific marker should be checked at least once, as described in ref 19.
  2. As author mentioned, EPC from ESKD patient showed low frequency and activity. This is why this study included data from 3-5 patients. During culture ex vivo, how about difference between health control and ESKD in aspects of cellular proliferation rate (represented as population doubling time; MTT assay represented the cellular status, not proliferation), cell morphology or cytokine secretion? The fundamental/cellular difference should be provided.
  3. How about the expression level of RXFP1 in EPC from the ESKD? This examined the RXFP1 on EPC from the healthy donor. This might determine the response of EPC to RLX.
  4. From author’s point, how can clinician apply this finding to ESKD sufferers? Should RLX is treated to patient with MSC therapy? The possibility for clinical application had better be described more detail.
  5. Please check panel lettering in Figure (C is missed in Figure 1)

Author Response

Biomedicines: Manuscript Ref 1672437R1

Response to Reviewers’ Comments: We thank the Reviewers for their constructive and insightful comments to improve the revised version of the manuscript. We have incorporated all the changes suggested by the reviewers and believe the revised manuscript is substantially improved and suitable for consideration of publication in Biomedicines. We have also submitted a red-marked version of the revised manuscript, so that the revisions made to the text can be easily identified.

Reviewers’ Comments:

Reviewer 1:

Comments and Suggestions for Authors

This study dealt with the critical effects of MSC/RLX on function of EPC that is impaired by donor disease. Considering the importance of circulating EPC in vasculogenesis, improvement of EPC activity is necessary in clinic. Author provided the data to support their hypothesis for combined effect of MSC-CM and RLX.

  1. Authors have cultured EPC and determined its type depending on culture period (Ref 23). Did author try to check stem cell marker expression? Although culture time is the decisive point to verify EPC type, cell-specific marker should be checked at least once, as described in ref 19.

         Authors’ Response: As previous studies in our laboratory had already reported on the stem cell marker expression of the EPC culture isolated, as described by Huuskes et al. (2018) in reference 19; in this study we did not investigate the stem cell marker expression of the EPC cultures. Furthermore, we no longer have access to the EPCs that were used in this study.

 2. As author mentioned, EPC from ESKD patient showed low frequency and activity. This is why this study included data from 3-5 patients. During culture ex vivo, how about difference between health control and ESKD in aspects of cellular proliferation rate (represented as population doubling time; MTT assay represented the cellular status, not proliferation), cell morphology or cytokine secretion? The fundamental/cellular difference should be provided.

Authors’ Response: The cellular morphology in this study was investigated using the colony forming unit assay, as described in lines 97-99 and in 213-216 of the revised manuscript. The number of colonies formed by ESKD patient-derived EPCs were significantly decreased when compared with EPCs derived from healthy controls.

3. How about the expression level of RXFP1 in EPC from the ESKD? This examined the RXFP1 on EPC from the healthy donor. This might determine the response of EPC to RLX.

Authors’ Response: As suggested by this Reviewer, we have examined RXFP1 expression in EPCs derived from the ESKD patients. In Figure 2A, we show the presence of immunoreactive RXFP1 in late EPCs derived from ESKD patients.

 4. From author’s point, how can clinician apply this finding to ESKD sufferers? Should RLX is treated to patient with MSC therapy? The possibility for clinical application had better be described more detail.

Authors’ Response: As suggested by this Reviewer, the clinical application of potential of the novel combination therapy reported in this study to enhance the regenerative potential of EPCs in ESKD patients is further detailed in lines 365-376, in the conclusion section of the revised manuscript.

 5. Please check panel lettering in Figure (C is missed in Figure 1).

Authors’ Response: We thank the Reviewer for highlighting the error in Figure 1. We have now inserted “C” in Figure 1.

Reviewer 2 Report

In this study the authors demonstrate that BM-MSC conditioned medium and serelaxin synergize to promote endothelial precursor cell (EPC) viability and angiogenic potential, both in normal EPCs from healthy donors and in impaired EPCs from in patients with end-stage kidney disease. The experiments have been carefully designed and performed. The results are clearly reported and properly discussed. The conclusions can have clinical relevance, which makes this manuscript suitable for the aims of Biomedicines journal. I have listed below some points requiring the authors’ attention.

Major points

1) Please remove the word ‘therapeutic’ from the title: this sounds as an over-statement since the results do not indicate any actual therapeutic effect: at most, they may suggest new future therapeutic perspectives.

2) Section 2.3.2: the MTT assay is primarily a viability assay since it gives a direct measure of mitochondrial redox capability: therefore, throughout the text and figure labels/legends, the terms ‘proliferation’ should be substituted for 'viability'. This does not substantially change the meaning of the experiments and the conclusions of the study. Anyhow, if the authors want to actually investigate cell proliferation, they could perform a specific assay, such as Ki67/MIB-1 immunolabeling (honestly, I feel no need for such supplement of investigation).

3) Section 2.3.6: please identify RXFP1 at its first mention as the specific H2 RLX receptor. Description of the immunofluorescence method is somewhat confusing: lane 180, what does ‘594 secondary antibody’ mean? I guess it is for ‘Alexa-fluor 594 nm-labelled secondary antibodies’; lane 182, DAPI is used for nuclear counterstaining, not for fluorescence detection (it is visualized ad a shorter excitation wavelength than Alexa fluorochromes).

4) Page 7, EPC cultures from ESKD patients: in this experimental set, the positive control with 20% FBS is missing, while it would have been important in order to assess the actual potential of these impaired EPCs (at least in the MTT viability assay).

5) Results, section 3.1: in my opinion, the demographic data of the subjects who donated the cells are unnecessary and not related to the reported in vitro results: these can be omitted, together with table 1.

6) Discussion: the first paragraph emphasizes some methodological aspects of EPC isolation and yield, while the reader would rather expect to begin with comments on the effects of the administered treatments, which is the key topic of this study. This paragraph should be moved to the end of this section and the discussion should begin with paragraph 2 ‘Previous studies…’.

Minor points

  • Please spell out ESKD at first mention in the abstract and in the title
  • Lane 131: ‘MTT labelling reagent was added and incubated’
  • Lane 155: 35 um or mm?
  • Lane 207: ‘per treatment group’
  • Figures 1 & 2: in all the bar graphs, the dots indicating individual values are redundant and rather mask the error bars: they can be omitted.
  • Lane 313: please add a sentence to summarize what Alport disease is.
  • Lane 346: please spell out AAD

Author Response

Reviewer 2:

Comments and Suggestions for Authors

In this study the authors demonstrate that BM-MSC conditioned medium and serelaxin synergize to promote endothelial precursor cell (EPC) viability and angiogenic potential, both in normal EPCs from healthy donors and in impaired EPCs from in patients with end-stage kidney disease. The experiments have been carefully designed and performed. The results are clearly reported and properly discussed. The conclusions can have clinical relevance, which makes this manuscript suitable for the aims of Biomedicines journal. I have listed below some points requiring the authors’ attention.

Major points

1) Please remove the word ‘therapeutic’ from the title: this sounds as an over-statement since the results do not indicate any actual therapeutic effect: at most, they may suggest new future therapeutic perspectives.

Authors’ Response: As suggested by this Reviewer we have now removed “therapeutic” from the title.

2) Section 2.3.2: the MTT assay is primarily a viability assay since it gives a direct measure of mitochondrial redox capability: therefore, throughout the text and figure labels/legends, the terms ‘proliferation’ should be substituted for 'viability'. This does not substantially change the meaning of the experiments and the conclusions of the study. Anyhow, if the authors want to actually investigate cell proliferation, they could perform a specific assay, such as Ki67/MIB-1 immunolabeling (honestly, I feel no need for such supplement of investigation).

Authors’ Response: The MTT assay used in this study was a commercially-purchased kit specifically designed for measuring both viability and proliferation according to the manufacturer’s recommendations (Sigma Aldrich/Merck). As suggested by this Reviewer we have now added viability to the functional analysis in lines 126, 127,236,254,270 and 353 of the revised manuscript.

3) Section 2.3.6: please identify RXFP1 at its first mention as the specific H2 RLX receptor. Description of the immunofluorescence method is somewhat confusing: lane 180, what does ‘594 secondary antibody’ mean? I guess it is for ‘Alexa-fluor 594 nm-labelled secondary antibodies’; lane 182, DAPI is used for nuclear counterstaining, not for fluorescence detection (it is visualized ad a shorter excitation wavelength than Alexa fluorochromes).

Authors’ Response: We acknowledge this comment from the Reviewer and in the revised manuscript, we have spelt out and identified RXFP1 at the start of Section 2.3.6; and added the correct descriptions for the Alexa Flour secondary and the nuclei detection by DAPI in lines 183 and 185.

4) Page 7, EPC cultures from ESKD patients: in this experimental set, the positive control with 20% FBS is missing, while it would have been important in order to assess the actual potential of these impaired EPCs (at least in the MTT viability assay).

Authors’ Response: We thank the Reviewer for pointing this out. We have now added the data from the effects of 20% FBS on ESKD patient-derived EPC viability/proliferation; which was found to restore the ESKD-induced loss of EPC viability back to levels measured from normal patients; and to a similar extent to 25% CM alone, RLX-10 alone or 25% CM+RLX-10 combined. In the revised manuscript, we have now added and discussed in lines 256 and 272, the effect of 20% FBS in the ESKD-derived EPCs.

5). Results, section 3.1: in my opinion, the demographic data of the subjects who donated the cells are unnecessary and not related to the reported in vitro results: these can be omitted, together with table 1.

Authors’ Response: We believe the demographic data is critical to inform both clinician researchers and scientists of the readership of Biomedicines about the descriptions and the nature of the patient samples used in this study. Therefore, we did not remove Table 1 from the revised manuscript.

6) Discussion: the first paragraph emphasizes some methodological aspects of EPC isolation and yield, while the reader would rather expect to begin with comments on the effects of the administered treatments, which is the key topic of this study. This paragraph should be moved to the end of this section and the discussion should begin with paragraph 2 ‘Previous studies…’.

Authors’ Response: As suggested by this Reviewer we have moved paragraph 2 to paragraph 1 of the Discussion.

Minor points

1) Please spell out ESKD at first mention in the abstract and in the title

Authors’ Response: We have spelt out ESKD as suggested by this reviewer in both the title (line 4) and in the abstract (line 17) of the revised manuscript.

2) Lane 131: ‘MTT labelling reagent was added and incubated’

Authors’ Response: We have corrected this statement in line 133 of the revised manuscript.

3) Lane 155: 35 um or mm?

Authors’ Response: We thank the reviewer for pointing out this error. In the revised manuscript we have corrected it to 35mm.

4) Lane 207: ‘per treatment group’

Authors’ Response: We have corrected this statement in line 210 of the revised manuscript as suggested by this reviewer.

5) Figures 1 & 2: in all the bar graphs, the dots indicating individual values are redundant and rather mask the error bars: they can be omitted.

Authors’ Response: We have not removed the dots as we believe it shows the biological variability of the samples used.

6) Lane 313: please add a sentence to summarize what Alport disease is.

Authors’ Response: We have described Alport disease line 322 of the revised manuscript.

7) Lane 346: please spell out AAD

Authors’ Response: We have spelt out AAD in line 350 of the revised manuscript as suggested by this reviewer.

Round 2

Reviewer 1 Report

Some issue was attempted to be resolved after revision.

Author Response

There were no specific comments provided by this reviewer.

Reviewer 2 Report

Just two final suggestions

1) lane 185-186, change to: "Nuclear counterstaining was performed by adding 40 µl of DAPI in Vectashield mounting medium'...

2) lane 322-323, change to: '(a genetic condition characterized by kidney microvascular abnormalities and functional failure)'

Author Response

We thank the reviewer for the final suggestions. We believe the revised manuscript is suitable for consideration for publication.

Comments and suggestions by Reviewer 2: 

1) lane 185-186, change to: "Nuclear counterstaining was performed by adding 40 µl of DAPI in Vectashield mounting medium'...

Authors' Response: As suggested by this reviewer, in the revised manuscript in line 185, we have now added "Nuclear counterstaining was performed by adding 40 µl of DAPI in Vectashield mounting medium".

2) lane 322-323, change to: '(a genetic condition characterized by kidney microvascular abnormalities and functional failure)'

Authors' Response: As suggested by this reviewer, in the revised manuscript in line 321, we have added (a genetic condition characterized by kidney microvascular abnormalities and functional failure).